# Caulerpin Mitigates *Helicobacter pylori*-Induced Inflammation via Formyl Peptide Receptors

**DOI:** 10.3390/ijms222313154

**Published:** 2021-12-05

**Authors:** Paola Cuomo, Chiara Medaglia, Ivana Allocca, Angela Michela Immacolata Montone, Fabrizia Guerra, Serena Cabaro, Ernesto Mollo, Daniela Eletto, Marina Papaianni, Rosanna Capparelli

**Affiliations:** 1Department of Agriculture, University of Naples Federico II, 80055 Naples, Italy; paola.cuomo@unina.it (P.C.); iv.allocca@gmail.com (I.A.); marina.papaianni@unina.it (M.P.); 2Department of Microbiology and Molecular Medicine, University of Geneva, Rue Michel-Servet 1, 1206 Geneva, Switzerland; chiara.medaglia@unige.ch; 3Department of Food Inspection, Istituto Zooprofilattico Sperimentale del Mezzogiorno, 80055 Naples, Italy; angela.montone@izsmportici.it; 4Department of Pharmacy, University of Naples Federico II, 80131 Naples, Italy; fabrizia.guerra@unina.it; 5Department of Translational Medicine, University of Naples Federico II, 80131 Naples, Italy; serena.cabaro@unina.it; 6National Research Council of Italy, Institute of Biomolecular Chemistry, 80078 Naples, Italy; emollo@icb.cnr.it; 7Department of Pharmacy, University of Salerno, Via Giovanni Paolo II 132, 84084 Salerno, Italy; daeletto@unisa.it; 8Task Force Microbioma, Department of Agriculture, University of Naples Federico II, 80055 Naples, Italy

**Keywords:** *Helicobacter pylori*, formyl peptide receptors, inflammation, bis-indole structure

## Abstract

The identification of novel strategies to control *Helicobacter pylori* (*Hp*)-associated chronic inflammation is, at present, a considerable challenge. Here, we attempt to combat this issue by modulating the innate immune response, targeting formyl peptide receptors (FPRs), G-protein coupled receptors that play key roles in both the regulation and the resolution of the innate inflammatory response. Specifically, we investigated, in vitro, whether Caulerpin—a bis-indole alkaloid isolated from algae of the genus *Caulerpa*—could act as a molecular antagonist scaffold of FPRs. We showed that Caulerpin significantly reduces the immune response against *Hp* culture filtrate, by reverting the FPR2-related signaling cascade and thus counteracting the inflammatory reaction triggered by *Hp* peptide Hp(2–20). Our study suggests Caulerpin to be a promising therapeutic or adjuvant agent for the attenuation of inflammation triggered by *Hp* infection, as well as its related adverse clinical outcomes.

## 1. Introduction

*Helicobacter pylori* (*H. pylori*) is a Gram-negative bacterium colonizing the gastric mucosa of over 50% of the humans worldwide [1]. Despite the fact that the infection is often asymptomatic, *H. pylori* represents one of the primary causes of gastric cancer, contributing to 75% of all gastric cancer cases [2,3]. While remaining a local pathogen, *H. pylori* may exert systemic effects and contribute to the occurrence of clinical extra-gastric manifestations. *H. pylori* infection, in fact, has been reported to increase the risk of iron deficiency anemia, as well as neurological, cardiovascular, dermatological and metabolic disorders [4,5,6,7]. The clinical outcomes of *H. pylori* infection depend on the complex relationship between host and bacterium [8,9]. The bacterium virulence, the host response, together with environmental factors contribute to the ability of *H. pylori* to colonize the harsh gastric environment over a long period, thus promoting long-term inflammation, a key feature for the development of severe gastric or extra-gastric diseases [10].

Inflammation is a defensive response triggered by the host innate immune system, in order to survive during an infection or an injury, favoring a return to homeostasis [11,12,13]. However, when prolonged, inflammation may cause more damage to the host than the pathogen [14], inducing intracellular metabolic changes and epigenetics modifications [15]. The immune system coordinates the inflammatory response through innate receptors, also known as pattern recognition receptors (PRRs), able to recognize highly conserved microbial structures [16].

Formyl peptide receptors (FPRs) are pattern recognition receptors (PRRs) belonging to the family of G_i_- protein coupled receptors, comprising FPR1, FPR2 and FPR3 [17,18]. Even though the nature of the evolutionary process responsible for FPRs’ differentiation is poorly understood, it is clear that it originates from a common ancestor and that they acquire functional differences through gene duplication and natural selection [19]. As PRRs, FPRs trigger cellular defense mechanisms, by sensing pathogen associated molecular patterns (PAMPs) [20]. Consequently, they play a critical role in host defense as well as in the regulation of inflammatory processes by participating in the pathogenesis of inflammatory disorders [21,22]. FPRs can detect both bacteria and host mitochondria-derived formylated peptides. In addition, FPRs, and specifically FPR2, can respond to a large variety of structurally different ligands, including not-formylated peptides.

Hp(2–20) is a *H. pylori*-derived not-formylated peptide [23]. More specifically, it is a cecropin-like peptide [23] showing broad-spectrum antimicrobial properties, which provide a competitive advantage over other microorganisms, thus facilitating *H. pylori* persistence in gastric mucosa [24]. Furthermore, as an exogenous ligand of FPR2, Hp(2–20) possesses pro-inflammatory activities [25], by initiating a cell-signaling cascade resulting in chemiotaxis, cytokines release and NADPH oxidase-dependent superoxide generation [23,26]. Both the antimicrobial, and the pro-inflammatory activity of Hp(2–20), makes it a significant *H. pylori* virulence factor, contributing to the pathogenesis of *H. pylori* infection and related chronic diseases. Importantly, by functioning in a ligand-specific manner, Hp(2–20) has been recognized as one of the primary risk factor for the development and progression of *H. pylori*-associated gastric cancer [23,27]. Given this scenario, FPRs play an important role in the severity of *H. pylori* infection, thus representing a potential target to control *H. pylori*-induced chronic inflammation and the associated risks.

The recent return to traditional medicine and natural drugs has increased the interest in marine natural products with potential pharmacological activity. Moreover, the current problem of marine biological invasion and their resulting biological, economic and social impact [28] have prompted the investigation of the biological activities of the natural products present in some of the most invasive species. In particular, *Caulerpa cylindracea*, a green macroalga native to South Western Australia and invasive in the Mediterranean Sea, has been found to contain high levels of the bis-indole alkaloid Caulerpin (Cau), showing anti-inflammatory and antioxidant activity [29,30,31]. The indole nucleus is a promising scaffold for the discovery of new anti-inflammatory and antinociceptive drugs, as it provides suitable ligands for G-protein coupled receptors [32,33].

We investigated, for the first time, whether Cau could exert an anti-inflammatory role in the context of *H. pylori* infection. We found that Cau inhibits FPR2, thus reverting the Hp(2–20)-induced signaling cascade. Our findings suggest a prospective therapeutic application of Cau as an adjuvant to control *H. pylori*-associated chronic inflammation and to prevent its related adverse effects. Nevertheless, based on the key role of FPRs in inflammatory disorders and of indole nucleus as “privileged structure” in drug discovery, Cau could represent a potential competitive alternative to classical anti-inflammatory approaches by preventing long-term inflammatory damage.

## 2. Results

### 2.1. The Effect of Cau on Cell Viability

First, we evaluated the effect of Cau on the metabolic activity of AGS gastric adenocarcinoma epithelial cells, using an MTT (3-4,5-dimethylthiazol 2,5-diphenyltetrazolium bromide solution) assay. AGS cells were cultured with different concentrations of Cau (spanning from 1.6 µM to 405 µM) for 24 h. As shown in Figure 1, Cau was not toxic up to 45 µM, since cell viability was higher than 80% compared to the untreated control. More specifically, we determined CC_50_ and observed that the concentration of Cau that caused a 50% decrease of cell viability was 61.20 µM.

### 2.2. Cau Reverses the Pro-Inflammatory Action of Hp(2–20)

Then, we assessed the anti-inflammatory effects of Cau by RT-qPCR, determining whether it could affect the induction of pro-inflammatory cytokines in response to *Helicobacter pylori* culture filtrate (Hpcf) or Hp(2–20). AGS cells were pretreated for 30 min with Cau with or without Hpcf or the *H. pylori* released peptide Hp(2–20), as indicated in Figure 2. Results (Figure 2A) showed that Cau alone does not induce the expression of *IL-6*, *IL-8* and *TNF*-*α* pro-inflammatory cytokines genes. On the contrary, both Hpcf and Hp(2–20) stimulated the expression of *IL-6*, *IL-8* and *TNF*-*α* genes, with a higher expression level for Hpcf. Interestingly, cells pretreated with Cau and then cultured with Hpcf or Hp(2–20) displayed a reduced expression level of *IL-6*, *IL-8* and *TNF*-*α*, compared to those not pretreated. Of note, the anti-inflammatory effect of Cau was stronger against Hp(2–20), rather than Hpcf. The data (Figure 2A,B) indicate that Cau exhibits in vitro anti-inflammatory properties. Further confirmation was provided by evaluating IkB-α phosphorylation (pIkB-α) through Western Blot analysis (Figure 2B). Results showed that Hp(2–20) and Hpcf increased the expression of pIkB-α in AGS cells, and, consistent with the above data, pIkB-α was higher in cells stimulated with Hpcf than Hp(2–20). Conversely, Cau pretreated cells displayed a reduced expression of pIkB-α, compared to those that were untreated and not pretreated (Figure 2B). Notably, the pIkB-α downregulation was higher in Hp(2–20)-stimulated cells than in those stimulated with Hpcf (Figure 2B). The appreciable pro-resolving effect of Cau in presence of the FPRs agonist Hp(2–20), suggests that the anti-inflammatory action of Cau is due to a direct interaction with FPRs. Finally, the demonstrable capability of Cau in inhibiting the expression level of *IL-8* gene (Figure 2A)—widely recognized as chemoattractant cytokine or chemokine—supports the hypothesis that Cau may control chemiotaxis and the oxidative burst [34] via FPRs.

### 2.3. The Effect of Cau on Hp(2–20)-Induced Chemotactic Signals

It has been reported that Hp(2–20) promotes the activation of chemotactic factors [35], thus triggering the FPRs downstream signaling pathway. Therefore, we investigated the effect of Cau on Hp(2–20)-induced chemotactic factors, by analyzing the expression profile of chemokine genes in AGS cells through RT-qPCR. As expected, Hp(2–20) induced a significant increase in chemotactic signals (Figure 3), while Cau reduced the expression of chemokine genes, similarly to the FPR2 antagonist WRW4, which we used as a positive control. Based on these findings, Cau, as well as WRW4, was found to inhibit FPR2 and the related signaling pathway. Furthermore, we performed an in vitro scratch assay. As expected, it confirmed the chemotactic activity of Hp(2–20) and interestingly, it revealed the capability of Cau to inhibit Hp(2–20)-induced cell migration (Appendix A), validating its potential role as an FPR2 antagonist.

### 2.4. Cau Prevents Hp(2–20)-Induced Inflammatory Response in Macrophages

Monocyte-derived macrophage recruitment is an important hallmark during inflammation [36]. According to different stimuli, macrophages change their polarization from M1 (classical activated macrophages) to M2 (alternatively activated macrophages) and vice versa, exhibiting either a pro-inflammatory or anti-inflammatory phenotype, respectively [37]. Previous results (Figure 2A and Figure 3) demonstrated that Hp(2–20)-stimulated cells up-regulated the expression of *CCL2*, *CCL3* and *TNF*-*α* genes, which are known to induce the macrophage pro-inflammatory phenotype [38,39]. Therefore, we evaluated the production of IL-1β and TNF-α—pro-inflammatory cytokines released by classically activated monocyte-derived macrophages—in THP-1 macrophages, cultured with the conditioned medium from AGS cells stimulated with Hp(2–20) either treated with Cau or left untreated. The THP-1 macrophages were incubated with the AGS conditioned medium for up to 24 h and their medium was collected at different time points, in order to measure the concentration of IL-1β and TNF-α using an ELISA assay, which normalized to the concentration of IL-1β and TNF-α contained in the conditioned medium of AGS cells that were differently stimulated. Figure 4 shows a time-dependent increase in IL-1β and TNF-α cytokines in cells treated with the conditioned medium of AGS stimulated with Hp(2–20) alone, and a marked decrease in the cytokines of cells treated with the conditioned medium of AGS stimulated with Hp(2–20) pretreated with Cau, suggesting the potential role of Cau in affecting the macrophage state.

### 2.5. Cau Affects the Hp(2–20)-Induced ROS Production via Mitochondrial and NADPH Oxidase-Dependent Mechanisms

Reactive oxygen species (ROS) play a critical role as microbicidal agents, participating in the initiation, as well as the resolution of the inflammatory process. However, if left uncontrolled, oxidative stress may be one of the primary causes of chronic inflammation and chronic inflammation-associated diseases, including cancer. To determinate whether Cau could affect Hp(2–20)-induced ROS production, AGS cells were pretreated with Cau and then stimulated with Hp(2–20), and intracellular ROS production was assessed. As shown in Figure 5, Hp(2–20) increased the ROS production compared to the untreated control, while Cau mitigated this effect. Interestingly, the Cau effect was comparable to the WRW4 response, suggesting that Cau reverses Hp(2–20)-stimulated ROS production by suppressing the FPR2 signaling cascade. Robust evidence has demonstrated that FPRs regulate oxidative burst via NADPH oxidase dependent ROS production [40]. Nevertheless, we also investigated the role of mitochondria in Hp(2–20)-induced ROS generation. Hp(2–20)-stimulated AGS cells were pretreated with rotenone (Rot), which inhibits the mitochondrial electron transport chain [41], causing a significant reduction in ROS production compared to cells stimulated with Hp(2–20) alone (Figure 6A). Rotenone reverted Hp(2–20)-induced ROS production similarly to WRW4. This suggests that mitochondria contribute to Hp(2–20)-induced ROS production. However, we observed that mitochondria-dependent ROS production upon Hp(2–20) stimulation was dose-dependent (Appendix A). This was further confirmed by examining the expression level of the mitochondrial antioxidant enzyme encoding gene *SOD2*, which was also found to be up-regulated in Hp(2–20) stimulated cells in a dose dependent manner (Appendix A) and importantly, it was found to be down-regulated in Cau pretreated cells (Figure 6B). Therefore, Cau hinders Hp(2–20)-induced ROS production acting on both mitochondria and NADPH oxidase via FPR2.

The pro-oxidant effect of Hp(2–20) was verified by evaluating, using RT-qPCR, the expression of the *p53* gene, which plays a central role in protecting cells from genetic insult by sensing cellular redox status [42]. In agreement with this finding, our results showed increased levels of *p53* gene in Hp(2–20) stimulated cells and a significant decrease in Cau pretreated cells (Figure 6C). Interestingly, p53 mRNA expression was found to decrease in a time-dependent manner in Hp(2–20) stimulated cells, with a marked decrease 6 h post stimulation (Figure 6C). Levels of p53 are, in fact, tightly regulated in response to an excessive levels of oxidative stress [43], as induced by the concentration of Hp(2–20) used, indicated through reduced levels of SOD2 gene compared to Hp(2–20), of 100 µM (Appendix A). This may explain why p53 gene levels decrease after the strong induction measured 1 h after Hp(2–20) treatment.

### 2.6. Cau Acts as FPR2 Inhibitor

To confirm whether Cau acts on FPR2, inhibiting Hp(2–20) downstream signaling, we first examined *FPR2* gene expression in response to Cau by performing quantitative real time PCR (qPCR). Figure 7A shows a similar trend in mRNA levels of FPR2 by WRW4 and Cau. Specifically, neither WRW4 nor Cau were found to activate FPR2. Interestingly, according to the classical receptor theory [44,45], they were found to up-regulate Hp(2–20)-induced *FPR2* gene expression over time, thus preventing FPR2 activation.

Furthermore, we also investigated the activity of Cau on FPR1 and FPR3. Our results showed a non-significant induction of FPR1 or FPR3 mRNA levels after 2 h of exposure to Hp(2–20) and consequently, a non-significant effect of both Cau and WRW4. These results suggest the specific action of Cau in inhibiting FPR2, the receptor showing higher affinity for Hp(2–20).

Next, we investigated the effect of Cau on the Hp(2–20)-induced signaling pathway. Upon Hp(2–20) binding, FPRs induced Akt, STAT3 and ERK1/2 phosphorylation [46]. Thus, we stimulated AGS cells—either with or without Cau pretreatment—with Hp(2–20) or Hpcf, and measured the STAT3, ERK1/2 and Akt phosphorylation by Western Blot analysis. As shown in Figure 7B, both Hp(2–20) and Hpcf-stimulated cells increased the STAT3, Akt and ERK1/2 phosphorylation compared to the control cells. Of note, ERK1/2 activation was higher in Hpcf-stimulated cells than in Hp(2–20)-stimulated cells, and, interestingly, ERK1/2 was found to be more phosphorylated in cells treated with Cau alone than in Hp(2–20)-stimulated cells (Figure 7B). This result is consistent with our idea to use Cau as a pro-resolving molecule, leading to gastric epithelium wound healing. Moreover, as expected, Cau pretreatment was found to reduce STAT3, Akt and ERK1/2 activation both in Hp(2–20) and Hpcf-stimulated cells (Figure 7B). However, Cau reduced ERK1/2 phosphorylation by only 10% and 30% in Hp(2–20) and Hpcf-stimulated cells, respectively. In addition, as well as ERK1/2, we observed a higher inhibition of Akt activation in Hpcf-stimulated cells rather than in Hp(2–20) ones. FPRs have been assumed to induce the transactivation of tyrosine kinase receptors (TKRs) via NADPH oxidase-dependent ROS production [18,46]. Previous results demonstrated the antioxidant effect of Cau by inhibiting FPR2 (Figure 5). However, it is well known that Cau is also able to scavenge free radicals in a receptor-independent manner [47]. Therefore, based on these findings, the major reduction of Akt and, to a lesser extent, ERK1/2 activation elicited by Cau in Hpcf-stimulated cells, may also be due to the effective antioxidant role of Cau, thus inhibiting TKRs interplay and the related signaling pathway.

### 2.7. Cau-FPR2 Interaction: Predictive Computational Studies

Finally, predictive molecular modeling studies were performed to investigate the interaction between Cau and FPR2. As shown in Figure 8A, Cau fits well with FPR2 ligand binding domain, occupying a small area of the receptor binding pocket. In addition, a partial overlap between WKYMV (FPR2 agonist) and Cau was predicted (Figure 8B). Cau was found to form hydrophobic interactions with FPR2 amino acids involved in hydrophobic interactions with WKYMV (Figure 8C). Specifically, Cau interacts with the amino acids (Phe257, Asp281, Asn285 and Arg201, Arg205) showing a critical role in ligand binding and formation of hydrogen bounds for FPR2 [48]. Despite the limits of this predictive approach, these data provide further evidence of the direct binding of Cau with FPR2.

### 2.8. Cau: A More Potent Anti-Inflammatory Molecule than Indomethacin

Indomethacin (Indo) is a member of the non-steroidal anti-inflammatory drugs (NSADs) class, used to treat inflammation and pain. Alongside the essential role of indomethacin in inhibiting prostaglandins synthesis, additional mechanisms could explain its potency. Both indomethacin and Cau possess the indole scaffold in their structure. More specifically, Cau presents an additional indole ring to indomethacin (Figure 9A). The structural similarity and the capability of indomethacin in suppressing formyl-peptides induced cell migration [49] led us to examine the potential interaction between indomethacin and FPRs. Indomethacin pretreatment, indeed, was found to modulate FPR2 induction triggered by Hp(2–20), as well as in cells pretreated with Cau (Figure 9B). These data provide evidence about the role of indomethacin on FPRs. We also observed a more potent anti-inflammatory effect for Cau than indomethacin. Figure 9C, shows a higher decrease in cytokines (MIP-1β, IL-8 and TNF-α) by Cau than indomethacin in cells stimulated with Hpcf.

## 3. Discussion

*Helicobacter pylori* is one of the most common human-colonizing bacteria, and its resultant infection can promote chronic inflammation. It is usually acquired during early childhood [50], remaining asymptomatic for long time. The significant capability of the bacterium in evading the host immune system and developing strategies to resist the common antimicrobial therapy means *H. pylori* is able to persist for decades in the harsh gastric environment, establishing lifelong chronic inflammation, which leads to severe clinical outcomes [1,9]. Strategies to control chronic inflammation, by modulating the immune system, may represent a promising approach to improve *H. pylori* clinical outcomes and counteract chronic diseases. 

In the present study, we focused our attention on FPRs, proposing FPRs as a novel target to ameliorate the detrimental effects derived from *H. pylori*-induced chronic inflammation. In particular, we investigated the potential capability of Cau to act as an attractive target for FPRs, inhibiting Hp(2–20) signaling pathway. The choice to use Cau in this study was based on its particular chemical structure, characterized by two indole nuclei. Indole, in fact, has been considered the most privileged scaffold in drug discovery [51,52] because of its anti-inflammatory, anti-cancer, antioxidant, anti-diabetic, antimicrobial, antiviral and anti-hypertensive roles [32,51,53]. In addition, the presence of two indole nuclei makes Cau similar to W-rich peptides, such as WRW_4_, which was found to interact with FPR2, exerting antagonistic effects [54].

For the first time, our results show the potential capability of Cau in antagonizing FPR2, as well as WRW_4_. Cau was tested as a probable target for all FPRs. However, it was found to distinguish between the three FPRs, interacting selectively with FPR2. The unique characteristic of indole rings in transferring electrons and favoring amino acids receptor reaction makes them effective components of the molecule. Specifically, Cau was observed to occupy the FPR2 binding pocket, forming a hydrophobic environment that could contribute to the stabilization of the receptor interaction, potentially competing with other ligands. This view was supported by the finding that Cau limited Hp(2–20)-induced cellular responses, via the inhibiting oxidative burst, chemotaxis and pro-inflammatory cytokines release.

NF-kB is the most important mediator of the inflammatory response, regulating the transcriptional activity of pro-inflammatory genes [55,56,57]. Our data, distinctly indicate an important role for Cau in suppressing Hp(2–20)-induced NF-kB signaling in gastric epithelial cells, by inhibiting phosphorylation and the degradation of the NF-kB inhibitor protein IkB-α. In contrast, Cau was not found to be as effective within the context of different bacterial stimuli. In the case of Hpcf, in fact, Cau did not display a significant suppression of IkB-α phosphorylation compared to when Hp(2–20) was used. To further assess the regulation of NF-kB pathway, *IL-8*, *IL-6* and *TNF-α* genes were selected to validate the above results. The potent pro-inflammatory cytokine, IL-8, plays a key role in initiating the chemotactic process and, together with IL-6 and TNF-*α,* is strongly associated with the *H. pylori*-related chronic inflammation [58,59]. Cau was found to reduce *IL-8, IL-6* and *TNF*-*α* gene transcription, both in Hp(2–20) and Hpcf stimulated cells. However, consistent with the previous data, it showed a more important inhibitory activity in Hp(2–20)-stimulated cells, than in Hpcf-stimulated cells. Taken together, these data indicate the pivotal role of Cau in modulating the inflammatory response, by acting selectively on FPRs. The modulation of Hp(2–20)-induced pro-inflammatory signaling by Cau was also demonstrated in monocyte-derived macrophages. Given the key role of macrophages in the immune response, we monitored the effect of Cau on macrophages exposed to Hp(2–20)-stimulated AGS cells microenvironment, observing a decreased expression of the pro-inflammatory cytokines IL-1β and TNF-α, commonly secreted by M1 polarized macrophages [60]. This suggests the importance of Cau as an FPR2 target, in mitigating the inflammatory response and affecting the state of macrophage polarization. The activation of gastric epithelial FPRs by Hp(2–20), triggers different signaling pathways, including MAPK/ERK, PI3K/Akt and STAT3 [23,61], which modulate important biological functions associated with the immune response—specifically, cell migration, proliferation and differentiation [27,62,63]. However, they also play central roles in tumor growth [62,64]. This makes Hp(2–20) controversial. A group of researchers demonstrated that Hp(2–20) induces gastric mucosal healing by stimulating cell migration and proliferation [61,65]. However, in the context of *H. pylori* persistence, the chronic activation of FPR2 by Hp(2–20) may promote cancer development and progression [27,66]. In agreement with the literature, our study indicated the increased activation of STAT3, Akt and ERK1/2 in Hp(2–20)-stimulated gastric epithelial cells, while Cau was found to reduce this response. Interestingly, the suppression of the above-mentioned key signaling molecules was found more remarkable in Hpcf-stimulated cells, than in those cultured with Hp(2–20). Apart from the FPRs signaling pathway, *H. pylori* activates different host-signaling pathways, responsible for cell and tissue alterations. Recent studies have demonstrated the critical role of the tyrosine kinase receptor (TKR) signaling pathway in *H. pylori* infection [67], as it leads to chronic inflammation and tumorigenesis [68,69]. It is well accepted that NADPH-derived ROS are highly involved in activating the TKRs signaling pathway and that FPRs induce TKRs transactivation via NADPH-oxidase dependent ROS production [18,70]. Here, we observed an increase in ROS production in Hp(2–20)-cultured cells, due to FPR2 activation and the important effect of Cau in neutralizing ROS by inhibiting FPR2 downstream signaling, as well as WRW4. Additional data demonstrated increased ROS production in Hpcf-stimulated cells than in Hp(2–20)-stimulated cells (data not shown), which was only partially attributable to FPR2 activation. Hpcf, contains many other factors which might contribute to ROS generation, thus causing TKRs signaling pathway activation. Therefore, the demonstrated antioxidant capability of Cau caused by inhibiting the FPRs signaling pathway, as well as its reported role as a free radical scavenger [47], could explain the unexpected beneficial effect of Cau in reducing STAT3, Akt and ERK1/2 phosphorylation, by inhibiting TKRs signaling pathway activation. This finding supports the success of Cau in counteracting *H. pylori*-related inflammatory diseases.

ROS are an important hallmark of inflammation. Excessive ROS generation is directly involved in the pathogenesis of several inflammatory disorders, including neurodegeneration, cardiovascular diseases, atherosclerosis and cancer [71,72]. Many studies have reported the crucial role of oxidative stress in the progression of *H. pylori*-related gastric carcinogenesis [73]. Moreover, the role of oxidative stress in the pathogenesis of *H. pylori*-associated extra-gastric diseases cannot be discounted [10]. FPRs are known to regulate the oxidative burst via NADPH oxidase-dependent ROS production [40]. However, studies have reported crosstalk between ROS produced at the mitochondria and the cytosol level, which exacerbates oxidative stress [74]. In the present study, we validated the pro-oxidant role of Hp(2–20) via FPRs and demonstrated the role of mitochondria in enhancing the oxidative stress from Hp(2–20). Interestingly, Cau was found to reverse Hp(2–20)-induced ROS production, targeting both mitochondria and NADPH oxidase.

Finally, we compared Cau with indomethacin, a traditional anti-inflammatory drug containing one indole nucleus. Extraordinarily, our data revealed that indomethacin may modulate the expression of FPR2, thus elucidating its potential pharmacological effect, which is still poorly understood. Nevertheless, Cau displayed a greater inhibitory effect against FPR2 than indomethacin and, at the same time, a more effective anti-inflammatory role. These results highlight the association between FPRs and inflammatory conditions and the importance of indole as a scaffold for anti-inflammatory drugs, by targeting FPRs.

## 4. Materials and Methods

### 4.1. Cell Culture

Human gastric AGS cell line and human monocytic THP-1 cell line were obtained from the American Type Culture Collection (ATCC, Manassas, VA, USA, #CRL-1739 and #TIB-202, respectively). AGS cells were grown in Dulbecco’s modification of Eagle’s medium, high glucose (DMEM; Microtech, Naples, Italy) supplemented with 10% fetal bovine serum (FBS; Microtech, Naples, Italy), 1% penicillin/streptomycin (Gibco, Waltham, MA, USA) and 1% L-glutamine (Gibco, Waltham, MA, USA). THP-1 cells were grown in RPMI-1640 (Microtech, Naples, Italy) and supplemented with 10% fetal bovine serum (FBS; Microtech, Naples, Italy), 1% penicillin/streptomycin (Gibco, Waltham, MA, USA) and 1% L-glutamine (Gibco, Waltham, MA, USA). Both AGS and THP-1 cell lines were maintained in a humidified environment, containing 5% CO_2_ at 37 °C. The THP-1 cells were induced to differentiate into macrophages through exposure to phorbol-12-myristate-13-acetate (PMA, 100 ng/mL; Sigma Aldrich, St. Louis, MO, USA) for 48 h. Cells were then washed twice, and culture medium was substituted with RPMI-1640 without PMA, followed by a resting period of 24 h.

### 4.2. Helicobacter Pylori Culture Filtrate Production

*H. pylori* P12 strains, kindly provided by Dr. Marguerite Clyne (University College Dublin), were cultured on selective Columbia agar (Oxoid, Basingstoke, Hampshire, UK) containing 7% (*v*/*v*) defibrinated horse blood (Oxoid, Basingstoke, Hampshire, UK) supplemented with an antibiotic mix (DENT or Skirrow, respectively, Oxoid, Basingstoke, Hampshire, UK). Bacteria plates were incubated for 3–4 days in a capnophilic atmosphere with 10% CO_2_ at 37 °C. Once grown on the plate, bacteria were scraped using brain heart infusion (BHI Oxoid, Basingstoke, Hampshire, UK) and measured at optical density at 600 nm (OD600) considering 1 OD600 = 1 × 10^8^ bacteria/mL. In order to prepare *H. pylori* broth growth, 2 × 10^7^ bacteria/mL were cultured in liquid DMEM (Euroclone, Milan, Italy), supplemented with 10% fetal bovine serum (FBS; Euroclone, Milan, Italy) and incubated in a capnophilic atmosphere with 10% CO_2_ at 37 °C. After 24 h, bacterial suspension was centrifuged at 10,000 g for 10 min to remove bacteria and the supernatant was filtered by using a 0.22 μm filter (Euroclone, Milan, Italy). The obtained culture filtrate was stored at −80 °C until use.

### 4.3. Cau Extraction and Purification

*Caulerpa cylindracea* was collected in Italy in the Gulf of Pozzuoli and exhaustively extracted with acetone at room temperature, as reported by Magliozzi et al. [75]. Briefly, the acetone extract was evaporated at a reduced pressure and the residual water was extracted with diethyl ether. The diethyl ether extract was first fractionated on Sephadex column (CHCl_3_/MeOH; 1:1) and the obtained fraction was further purified by silica-gel column chromatography (gradient of light petroleum ether/Et_2_O, as eluent) to produce pure Cau, identified by comparing ^1^H- and ^13^C-NMR spectroscopic data with the literature [76,77]. Size-exclusion chromatography was achieved using Sephadex LH-20 column, whereas silica-gel column chromatography was performed using Merck Kieselgel 60 powder. NMR data were recorded on a Bruker Avance-400 spectrometer using an inverse probe fitted with a gradient along the z axis.

### 4.4. Cell Viability Assay

The effects of Cau on AGS cells were assessed by performing MTT assay. Briefly, AGS cells were seeded at a density of 2 × 10^3^ per well in a 96-well plate and incubated at 37 °C in a 5% CO_2_ atmosphere overnight. After cell attachment, the medium was replaced with fresh medium containing different concentrations of Cau and cells were incubated for 24 h. Twenty µL of 3-4,5-dimethylthiazol 2,5-diphenyltetrazolium bromide solution (MTT) were added to each well and cells were further incubated at 37 °C in a 5% CO_2_ atmosphere for 3 h. Finally, the medium was removed, and the resultant formazan crystals were dissolved in 200 µL of DMSO. Absorbance was recorded at 570 nm using an EnVision 2102 multilabel reader (PerkinElmer, Waltham, MA, USA). Cell viability was calculated as the ratio between the mean absorbance of the sample and the mean absorbance of the untreated cells and expressed in percentage.

### 4.5. In Vitro Scratch Assay

AGS cells migration was tested by performing in vitro scratch assay, as described by de Paulis et al. [61]. Briefly, confluent monolayers of cells were treated with mitomycin for 2 h (2 µg/mL) to inhibit cell growth. Monolayers were then scratched using a pipette tip, in order to create a gap. After scratching, medium and cell debris were removed, and cells were washed with a fresh medium and incubated for 12 h with Hp(2–20) with or without Cau pretreatment.

### 4.6. RNA Extraction and Quantitative Real-Time PCR

AGS cells were seeded at a density of 0.5 × 10^6^ per well in 12-well plates to analyze the expression profiling of (1) cytokine and chemokine genes; (2) *FPRs* genes; (3) mitochondrial superoxide dismutase (*SOD2*) gene and (4) *p53* gene at different times post-inflammatory stimulus, represented by Hp(2–20) (synthesized by Innovagen, Lund, Sweden) or Hpcf, in the presence or absence of the Cau pretreatment. Total RNA was extracted from individual wells by PureLink^®^ RNA Mini Kit (ThermoFisher Scientific, Waltham, MA, USA), according to the manufacturer instructions. Genomic DNA was removed by digestion with DNase I, Amplification Grade (TermoFisher Scientific). Extracted RNA was quantified and analyzed for purity, using Nanodrop-ND 1000 spectrophotometer (TermoFisher Scientific) and finally reverse-transcribed using the high-capacity cDNA Reverse transcription kit (Applied Biosystem, Bedford, MA, USA). Gene transcript levels were measured using TaqMan PCR master 2× reagent or Power SYBR^®^ Green PCR Master Mix (Applied Biosystem^®^) on a StepOne^™^ Real-Time PCR System (Applied Biosystem^®^), according to the standard-mode thermal cycling conditions, according to the manufacturer’s protocol. The relative expression level of analyzed genes was determined using probes or primers reported in the Appendix A. All samples were normalized to GAPDH as the reference housekeeping gene and the relative quantitative expression was determined using the 2^−^^ΔΔCt^ method [78,79,80].

### 4.7. Measurement of Cytokines Production in Macrophages

Opportunely differentiated THP-1 cells were seeded at a density of 0.5 × 10^6^ per well in 12-well plates and stimulated with AGS-conditioned medium for 24 h. AGS-conditioned medium was prepared by seeding 2 × 10^6^ cells per well in 6-well plates. After treatment with the pro-inflammatory stimulus Hp(2–20), preceded or not by Cau pre-treatment, the medium was collected and filtered by passage in a 0.22 μm filter (Sigma Aldrich). Supernatants of THP-1 cells were collected at different times post-stimulation with AGS-conditioned medium and stored at −80 °C until use. Secretion of IL-1β and TNF-α was detected by Human ELISA kit (Abcam, Waltham, MA, USA), according to the manufacturer instructions.

### 4.8. Intracellular ROS Measurement

AGS cells were split at 80–90% of confluency, seeded (0.5 × 10^6^) in 35 mm culture dishes, and incubated at 37 °C in a 5% CO_2_ atmosphere overnight. After cell attachment, ROS detection assay was assayed using dihydrorhodamine 123 (DHR; Sigma Aldrich, Missouri, USA), as described by Cuomo et al. [81]. Briefly, cells were preloaded with 10 μM DHR for 20 min and treated as detailed in the figure legend. After treatments, DAPI (Thermo Fischer Scientific) was used as a nuclear counterstain, and cells were analyzed with a Zeiss Axioskop 2 Hal100 fluorescence microscope equipped with a digital camera (Nikon). The excitation and emission wavelengths were 488 and 515 nm, respectively. Images were digitally acquired with exposure times of 100–400 ms and processed for fluorescence determination with ImageJ software version 2.1.0/1.53c.

### 4.9. Western Blotting for Protein Studies

AGS cells were split at 80–90% of confluency, seeded (2 × 10^6^ per well) in 6-well plates and incubated at 37 °C in a 5% CO_2_ atmosphere until cell attachment. Cells were serum-starved 12–16 h prior the stimulation, using serum-free DMEM containing 0.25% BSA and incubated at 37 °C in a CO_2_ incubator. After treatments, cells were washed and harvested using RIPA buffer (50 mM Tris-HCl pH 8.8, 150 mM NaCl, 0.1% SDS, 0.5% NP-40, 0.5% DOC; protease and phosphatase inhibitor cocktail, Sigma-Aldrich), incubated for 20 min at 4 °C and centrifugated at 10,000× *g* for 15 min. Pellets were discarded and cell lysates were stored at −80 °C until use. Proteins (30 μg/lane, in Sample Buffer: 4× Laemmli Sample Buffer #1610747) were separated in 7.5–15% SDS-polyacrylamide gel and then transferred to a nitrocellulose membrane (Amersham™ Nitrocellulose Western blotting membranes 0.2um) by electrotransfer. Briefly, filters were blocked for 1 h at room temperature in 5% (*w*/*v*) non-fat milk in Tris-buffered saline Tween-20 (TBST: 0.1% Tween, 150 mM NaCl, 10 mM Tris-HCl, pH 7.5) and probed with antibodies as reported in the Appendix A. After several washings in TBST, membranes were incubated with the appropriate secondary antibodies (Appendix A). Finally, immunoreactive proteins were visualized with enhanced chemiluminescence (Amersham International, Buckinghamshire, UK). The blots were stripped using a stripping buffer and then re-probed to detect the total protein of interest. Each Western blot band was quantified using Image Lab software version 6.0 (Bio-Rad, Hercules, CA, USA).

### 4.10. Cytokines Assay

AGS cells were split at 80–90% of confluency, seeded (2 × 10^6^ per well) in 6-well plates and incubated at 37 °C in a 5% CO_2_ atmosphere until cell attachment. Cells were serum-starved 12–16 h prior the stimulation, using serum-free DMEM containing 0.25% BSA and incubated at 37 °C in a CO_2_ incubator. After 24 h of treatment, medium was collected and centrifugated at 10,000× *g* to remove debris and dead cells and analysed for the concentration of IL-8, G-CSF, TNF-α and MIP-1β, by using the Bioplex Multiplex human cytokine assay (Bio-Rad), as indicated by manufacturer’s instructions. As a multiplexed assay, the Bioplex assay can simultaneously detect more analytes in a single sample [82].

### 4.11. Molecular Modeling

To predict and characterize the potential interaction of Cau with FPR2, computational studies were assessed. A 3D structure model of FPR2 was downloaded from Protein Data Bank (RCSB PDB), using the crystal structure of the complex FPR2-WKYMVm (PDBcode: 6LW5) as a model. The WKYMV ligand was successively removed by PyMOL Molecular Graphic System (Version 1.3 Shrodinger, LLC, New York, NY, USA), obtaining the receptor pdb file. The initial 3D conformation of Cau was obtained from PubChem (ncbi library). The receptor and ligand were then adapted for docking with the AutoDock Tools. Docking analysis was carried out with AutoDock Vina (Trott and Olson, 2010), setting the grid box at 26Å × 40Å × 32Å for the receptor. In conclusion, molecular details of Cau recognition by FPR2 were analyzed with PyMOL Molecular Graphic System (Version 1.3 Shrodinger, LLC).

### 4.12. Statistical Analysis

Statistical analysis was performed using GraphPad Prism 8.0 software, San Diego, CA, USA. All data were compared using One-way ANOVA followed by Bonferroni’s multiple comparisons test, in order to compare different groups. Experimental data are presented as mean ± SD of at least three independent experiments, each performed in triplicate. Lastly, *p* values < 0.05 were considered statistically significant.

## 5. Conclusions

In conclusion, the current study aimed to provide in vitro proofs on the role of FPRs in the pathogenesis of *H. pylori*-associated chronic inflammation, by interacting with the *H. pylori*-released peptide Hp(2–20). Concurrently, it demonstrated the impressive effects of Cau on health, by targeting FPR2, thus controlling the *H. pylori*-associated chronic inflammation and related disorders. Taken together, our results suggest the potential clinical application of Cau for the control of numerous inflammatory disorders, which are among the main health problems occurring today. Nevertheless, future studies are required to validate our in vitro findings in vivo.

## Figures and Tables

**Figure 1 ijms-22-13154-f001:**
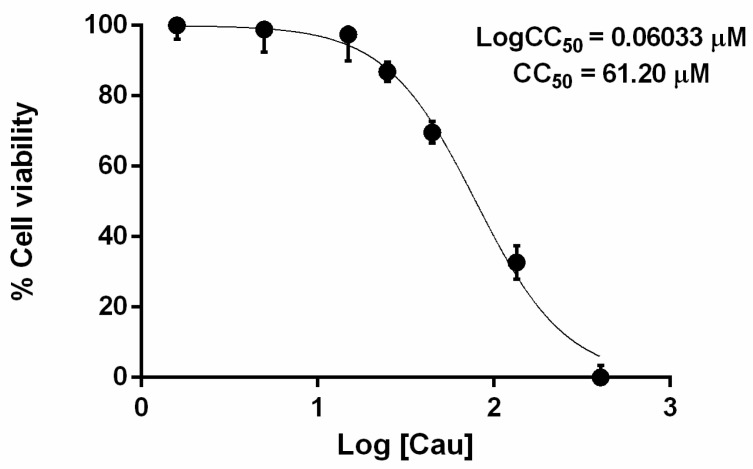
Effect of Cau on AGS cell viability. AGS cells were cultured with different concentrations of Cau for 24 h. Results were obtained by combining three independent experiments and represented as mean ± SD compared to control cells (100% viability).

**Figure 2 ijms-22-13154-f002:**
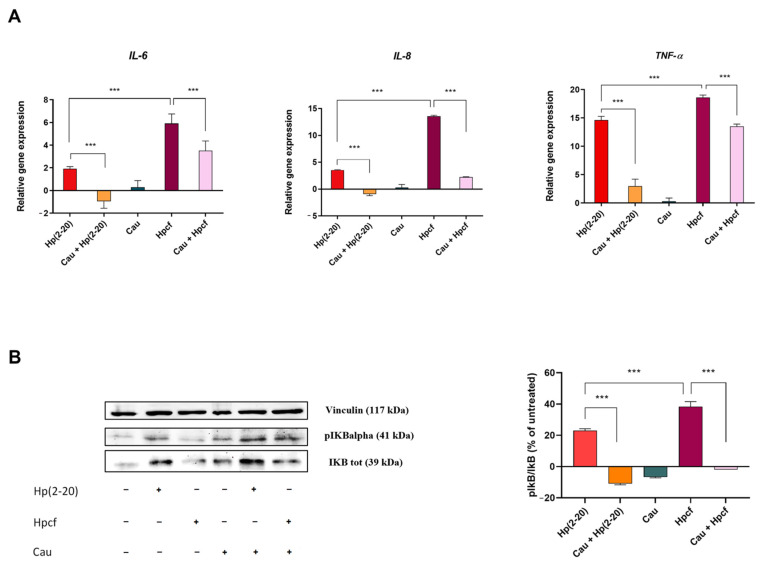
Cau significantly decreases the in vitro *H. pylori*-induced inflammatory response. (**A**) Relative gene expression of *IL-6*, *IL-8* and *TNF*-α was determined by quantitative real time PCR (qPCR), performed on RNA isolated from AGS cells cultured with Hp(2–20) 25 µM or Hpcf for 2 h and cells pretreated with Cau 15 µM and then cultured with Hp(2–20) 25 µM or Hpcf for 2 h. All samples were normalized to GAPDH as the reference housekeeping gene. Furthermore, relative gene expression was normalized to basal activity (untreated control), in order to obtain relative fold expression. (**B**) Western Blot and densitometric analysis of the ratio pIkB-α/IkB-α, compared to control group. The analysis was performed on cell lysate prepared after stimulation with Hp(2–20) or Hpcf for 1 h with or without Cau pretreatment (30 min). Vinculin was used for normalization. Graphs report the results of at least three independent experiments, represented as means ± SD. Statistical analysis was performed by GraphPad Prisma software, using one-way ANOVA followed by Bonferroni post hoc correction. *** *p* < 0.0001.

**Figure 3 ijms-22-13154-f003:**
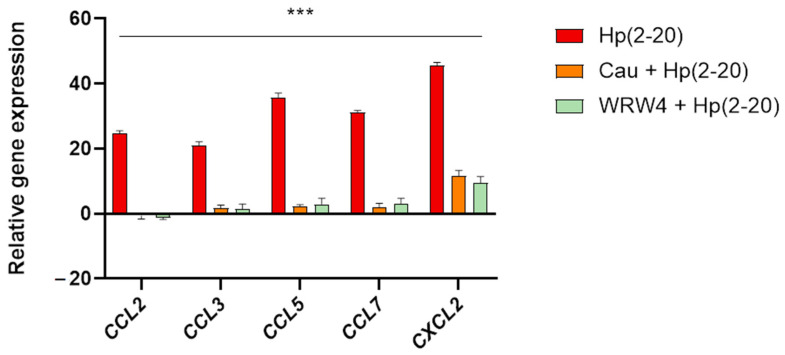
Cau attenuates Hp(2–20)-induced chemotactic signals. Relative genes expression of *CCL2*, *CCL3*, *CCL5*, *CLL7* and *CXCL2* was determined by quantitative real time PCR (qPCR), performed on RNA isolated from AGS cells cultured with Hp(2–20) 25 µM for 2 h and cells pretreated with Cau 15 µM or WRW4 10 µM for 30 or 15 min, respectively, and then cultured with Hp(2–20) 25 µM for 2 h. All samples were normalized to GAPDH as the reference housekeeping gene. Furthermore, relative gene expression was normalized to basal activity (untreated control), in order to obtain relative fold expression. Graphs report the results of at least three independent experiments, represented as means ± SD. Statistical analysis was performed by GraphPad Prisma software, using one-way ANOVA followed by Bonferroni post hoc correction. *** *p* < 0.0001 (Hp(2–20) vs. Cau + Hp(2–20) and Hp(2–20) vs. WRW4 + Hp(2–20).

**Figure 4 ijms-22-13154-f004:**
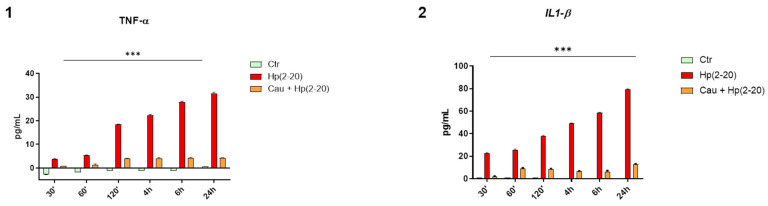
Cau prevents AGS-mediated macrophages activation. Cytokines: (**1**) TNF-α and (**2**) IL-1β were measured by performing ELISA in macrophages culture medium, collected at different times. Results are expressed as pg of cytokines secreted in mL of macrophages medium, differently treated: (1) conditioned medium of AGS cells cultured with Hp(2–20) 25 µM for 24 h; (2) conditioned medium of AGS cells pretreated with Cau 15 µM for 30 min and then cultured with Hp(2–20) 25 µM for 24 h. Values were normalized to the concentration of the cytokines of the AGS conditioned medium and represent mean ± SD of at least three independent experiments, each performed in triplicate. Statistical analysis was performed by GraphPad Prisma software, using one-way ANOVA test followed by Bonferroni post hoc correction. *** *p* < 0.0001 (Ctr vs. Hp(2–20) and Hp(2–20) vs. Cau + Hp(2–20)).

**Figure 5 ijms-22-13154-f005:**
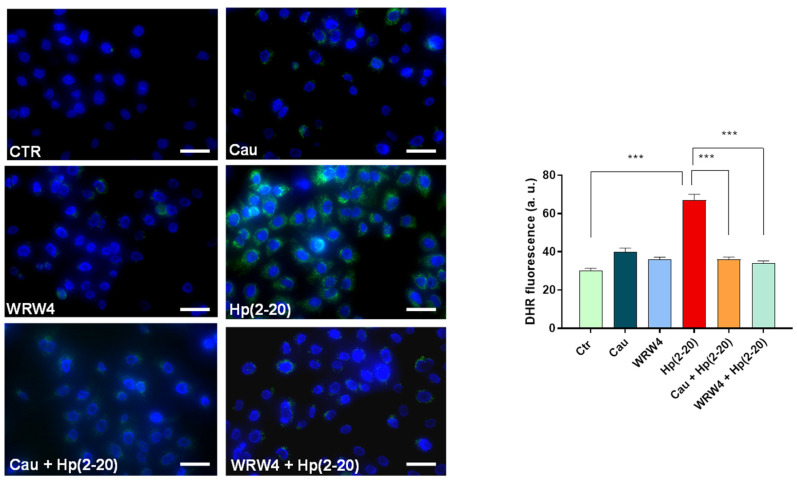
Cau moderates FPR2-mediated ROS production. DHR-loaded cells differently treated: (1) control cells; (2) Hp(2–20) 25 μM for 1 h; (3) Cau 15 μM for 1.5 h; (4) WRW4 10 μM for 1.5 h; (5) Cau 15 μM for 30 min and Hp(2–20) 25 μM for 1 h; (6) WRW4 10 μM for 15 min and Hp(2–20) 25 μM for 1 h, observed with a fluorescence microscope (on the left, scale bar: 100 μm) and quantification of the mean fluorescence of individual cells (on the right). Results are expressed as arbitrary units and represent the mean ± SD calculated from three independent experiments, each performed in duplicate. Statistical analysis was performed by GraphPad Prisma software, using one-way ANOVA followed by Bonferroni post hoc correction. *** *p* < 0.0001.

**Figure 6 ijms-22-13154-f006:**
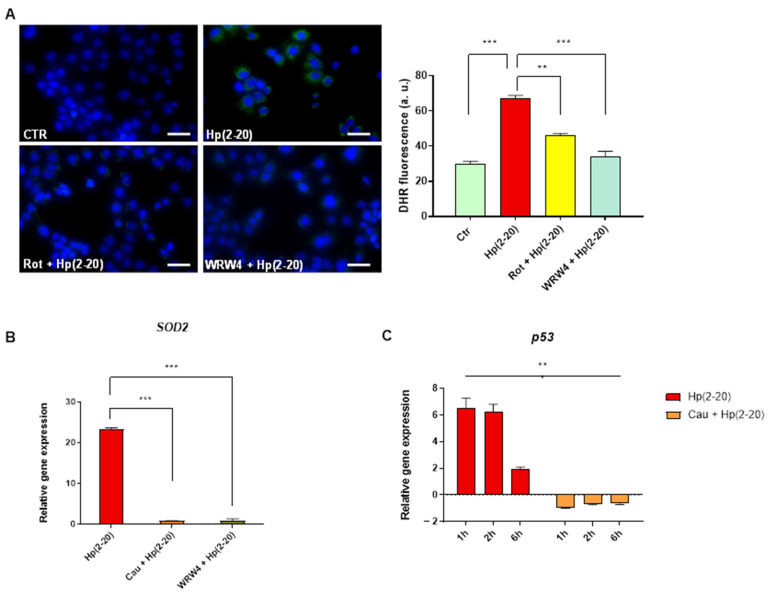
Cau reverses Hp(2–20)-induced ROS produced by mitochondria. (**A**) DHR-loaded cells differently treated: (1) control cells; (2) Hp(2–20) 100 μM for 1 h; (3) Rotenone 0.5 mM for 15 min and Hp(2–20) 100 μM for 1 h; (4) WRW4 10 μM for 15 min and Hp(2–20) 100 μM for 1 h, observed with a fluorescence microscope (on the left, scale bar: 100 μm) and quantification of the mean fluorescence of individual cells (on the right). Results are expressed as arbitrary units and represent the average ± SD calculated from three independent experiments each performed in duplicate. Statistical analysis was performed by GraphPad Prisma software, using one-way ANOVA. ** *p* < 0.001; *** *p* < 0.0001. (**B**) Relative gene expression of *SOD2* by quantitative real time PCR (qPCR) in cells treated with: (1) Hp(2–20) 25 μM for 2 h; (2) Cau 15 μM for 30 min and then Hp(2–20) 25 μM for 2 h; (3) WRW4 10 μM for 15 min and then Hp(2–20) 25 μM for 2 h. (**C**) Relative gene expression of *p53* by quantitative real time PCR (qPCR) in cells treated with (1) Hp(2–20) 25 μM for 1 h; (2) Hp(2–20) 25 μM for 2 h; (3) Hp(2–20) 25 μM for 6 h; (4) Cau 15 μM for 30 min and Hp(2–20) 25 μM for 1 h; (5) Cau 15 μM for 30 min and Hp(2–20) 25 μM for 2 h; (6) Cau 15 μM for 30 min and Hp(2–20) 25 μM for 6 h. All samples were normalized to GAPDH as reference housekeeping gene. Furthermore, relative gene expression was normalized to basal activity (untreated control), in order to obtain relative fold expression. Graphs report the results of at least three independent experiments, represented as means ± SD. Statistical analysis was performed by GraphPad Prisma software, using one-way ANOVA followed by Bonferroni post hoc for multi-comparison (more than two groups) or student’s t test for single-comparison (two groups). ** *p* < 0.001; *** *p* < 0.0001.

**Figure 7 ijms-22-13154-f007:**
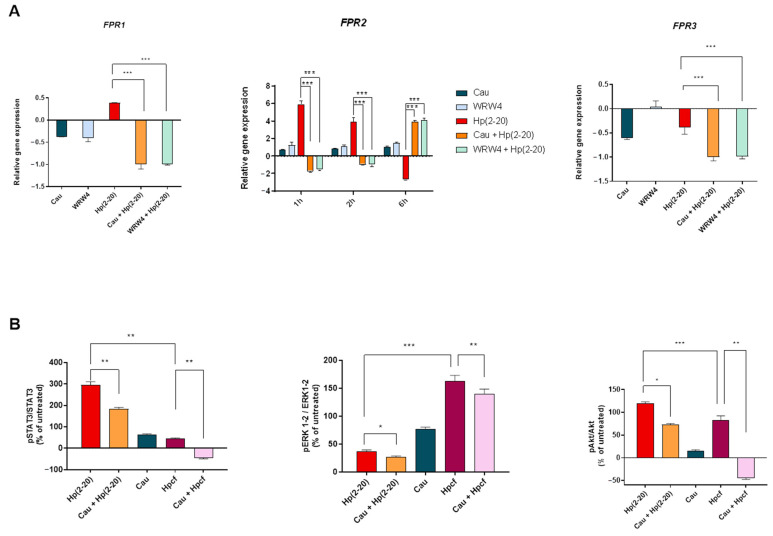
Cau inhibits FPR2 and its signaling cascade. (**A**) Relative gene expression of *FPR1*, *FPR2* and *FPR3* by quantitative real time PCR (qPCR) in cells treated with: (1) Cau 15 μM for 2.5 h; (2) WRW4 10 μM for 2.25 h; (3) Hp(2–20) 25 μM for 2 h; (4) Cau 15 μM for 30 min and then Hp(2–20) 25 μM for 2 h; (5) WRW4 10 μM for 15 min and then Hp(2–20) 25 μM for 2 h. *FPR2* gene was further analyzed in cells treated as reported above, but incubated with Hp(2–20) for 1 or 6 h. All samples were normalized to GAPDH as reference housekeeping gene. Furthermore, relative gene expression was normalized to basal activity (untreated control), in order to obtain relative fold expression. (**B**) Densitometric analysis derived from Western Blot of the ratio pSTAT3/STAT3; pERK1-2/ERK1-2; pAkt/Akt, compared to control group. The analysis was performed on cell lysate prepared after stimulation with Hp(2–20) or Hpcf for 1 h; 15 or 5 min, respectively, either with or without Cau pretreatment (30 min). Vinculin was used for normalization. Graphs report the results of at least three independent experiments, represented as means ± SD. Statistical analysis was performed by GraphPad Prisma software, using one-way ANOVA followed by Bonferroni post hoc correction. * *p* < 0.05; ** *p* < 0.001; *** *p* < 0.0001.

**Figure 8 ijms-22-13154-f008:**
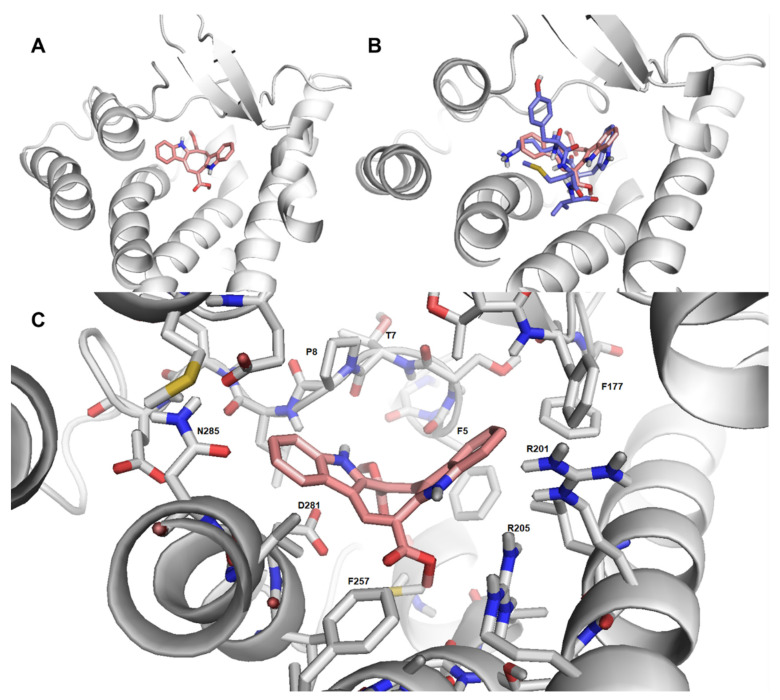
Cau interacts with FPR2, occupying its binding pocket. (**A**) Schematic representation of the complex FPR2-Cau. The receptor is represented in grey cartoon, while the ligand is shown in pink sticks. (**B**) Overall structure of the complex FPR2-Cau-WKYMV. The receptor is shown in grey cartoon, while the ligands are shown in pink or blue sticks, respectively. A partial overlap between Cau and WKYMV is shown. (**C**) Binding pocket of Cau in FPR2. The receptor is represented in grey cartoon, while the ligand is shown in pink sticks. The amino acids of the binding site are represented via grey sticks. Cau forms hydrophobic interaction with amino acids of the extracellular domain, I; II; III extracellular portion and VI transmembrane portion, which also constitute the WKYMV binding pocket.

**Figure 9 ijms-22-13154-f009:**
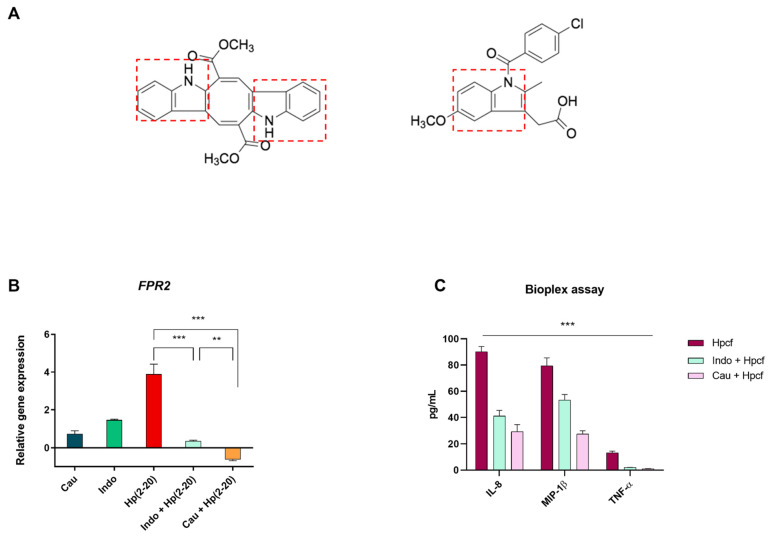
Cau possesses one additional indole nucleus to indomethacin and therefore major anti-inflammatory effects. (**A**) Molecular structure of Cau (on the left) is compared to indomethacin (on the right). Both Cau and indomethacin contain an indole scaffold, indicated with a red square. (**B**) Relative gene expression of *FPR2* by quantitative real time PCR (qPCR) in cells treated with: (1) Cau 15 μM for 2.5 h; (2) WRW4 10 μM for 2.25 h; (3) Hp(2–20) 25 μM for 2 h; (4) Cau 15 μM for 30 min and then Hp(2–20) 25 μM for 2 h; (5) Indomethacin100 μg/mL for 30 min and then Hp(2–20) 25 μM for 2 h; (6) Hpcf; (7) Indomethacin100 μg/mL for 30 min and then Hpcf for 2 h; (8) Cau 15 μM for 30 min and then Hpcf for 2 h. All samples were normalized to GAPDH as the reference housekeeping gene. Furthermore, relative gene expression was normalized to basal activity (untreated control), in order to obtain relative fold expression. Graphs report the results of at least three independent experiments, represented as means ± SD. (**C**) Cytokines TNF-α, IL-8 and MIP-1β were measured by Bio-plex assay in AGS cells culture medium. Results are expressed as pg of cytokines secreted in mL of cell medium, differently treated: (1) Hpcf for 24 h; (2) Cau 15 µM for 30 min and then Hpcf for 24 h; (3) indomethacin 200 µM for 30 min and then *H. pylori* culture supernatant for 24 h. Values were normalized to basal activity (CTR) and represent mean ± SD of at least three independent experiments, each performed in triplicate. Statistical analysis was performed by GraphPad Prisma software, using one-way ANOVA followed by Bonferroni post hoc correction. ** *p* < 0.001; *** *p* < 0.0001.

## Data Availability

The data presented in this study are available within the article.

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
