# Peer review of "Caulerpin Mitigates Helicobacter pylori-Induced Inflammation via Formyl Peptide Receptors"

_ijms, 2021, doi:10.3390/ijms222313154_

Round 1

Reviewer 1 Report

Cuomo’s study investigated whether a formyl peptide receptor antagonist Caulerpin could reduced the immune response against HP infection. They showed that Caulerpin significantly reverted the FPR2-relateed signaling and thus reduced the inflammation. Their study suggested that Caulerpin could be a potential agent for controlling chronic inflammation triggered by HP infection. This study is interesting and might be important for clinical therapies against HP. However, several questions have to be addressed.

  1. The authors tested the effect of Cau mainly by in vitro experiments based on cells. It would be better if the authors could address the same questions in H. pylori-infected animal models.
  2. The authors used Hp(2-20) to mimic the effects of Hp. However, the T4SS system of Hp is very important for its role in causing inflammation. It is better for the author to use alive Hp strains, such as PMSS1, to confirm their results.
  3. Figure 2B , is pIKBalpha normalized to Vinculin rather than total IKB? It seems that the expression of pIKB in the left panel is not correlated with statistic results in the right panel, especially the Hpcf.
  4. Figure 6C, Bonferroni is used for multi-comparision. There is only one comparison result was shown in figure 6C ,why should the author use Bonferroni. Besides, p53 is a unstable protein, and its protein level is always not correlated with the mRNA level, could the author show the protein level of p53?

Reviewer 2 Report

The manuscript is very interesting and sounding, reporting new findings about natural products effective against H. pylori.

The study is well designed and well conducted, statistical methods and analysis are correct. Discussion and conclusion are adequate.

I've only a concern regarding the Introduction. The authors report (lines 35-36) that Hp is one of the causes of gastric cancer, but they should report that it accounts for other pathologies related to a chronic or persistent inflammatory status. In fact, gastric MALTomas, coronaric diseases, skin conditions, anemia, etc.

I suggest to briefly discuss such conditions, also considering the following papers:

  • Charitos, I.A.; D’Agostino, D.; Topi, S.; Bottalico, L. 40 Years of Helicobacter pylori: A Revolution in Biomedical Thought. Gastroenterol. Insights. 2021, 12, 111-135 doi: 10.3390/gastroent12020011
  •  Durazzo M, Adriani A, Fagoonee S, Saracco GM, Pellicano R. Helicobacter pylori and Respiratory Diseases: 2021 Update. Microorganisms. 2021 Sep 26;9(10):2033. doi: 10.3390/microorganisms9102033. 
  • Kiesewetter B, Simonitsch-Klupp I, Mayerhoefer ME, Dolak W, Lukas J, Raderer M. First Line Systemic Treatment for MALT Lymphoma-Do We Still Need Chemotherapy? Real World Data from the Medical University Vienna. Cancers (Basel). 2020 Nov 26;12(12):3533. doi: 10.3390/cancers12123533. 
  • Mladenova I. Helicobacter pylori and cardiovascular disease: update 2019. Minerva Cardioangiol. 2019 Oct;67(5):425-432. doi: 10.23736/S0026-4725.19.04986-7
  • Shindler-Itskovitch T, Chodick G, Shalev V, Muhsen K. Helicobacter pylori infection and prevalence of stroke. Helicobacter. 2019 Feb;24(1):e12553. doi: 10.1111/hel.12553. 
  • Li JZ, Li JY, Wu TF, Xu JH, Huang CZ, Cheng D, Chen QK, Yu T. Helicobacter pylori Infection Is Associated with Type 2 Diabetes, Not Type 1 Diabetes: An Updated Meta-Analysis. Gastroenterol Res Pract. 2017;2017:5715403. doi: 10.1155/2017/5715403.

Don't use italics for cytokines' names.

Author Response

Response to Reviewer 2 Comment

The manuscript is very interesting and sounding, reporting new findings about natural products effective against H. pylori.

The study is well designed and well conducted, statistical methods and analysis are correct. Discussion and conclusion are adequate.

I've only a concern regarding the Introduction. The authors report (lines 35-36) that Hp is one of the causes of gastric cancer, but they should report that it accounts for other pathologies related to a chronic or persistent inflammatory status. In fact, gastric MALTomas, coronaric diseases, skin conditions, anemia, etc.

I suggest to briefly discuss such conditions, also considering the following papers:

  • Charitos, I.A.; D’Agostino, D.; Topi, S.; Bottalico, L. 40 Years of Helicobacter pylori: A Revolution in Biomedical Thought. Gastroenterol. Insights. 2021, 12, 111-135 doi: 10.3390/gastroent12020011
  •  Durazzo M, Adriani A, Fagoonee S, Saracco GM, Pellicano R. Helicobacter pylori and Respiratory Diseases: 2021 Update. Microorganisms. 2021 Sep 26;9(10):2033. doi: 10.3390/microorganisms9102033. 
  • Kiesewetter B, Simonitsch-Klupp I, Mayerhoefer ME, Dolak W, Lukas J, Raderer M. First Line Systemic Treatment for MALT Lymphoma-Do We Still Need Chemotherapy? Real World Data from the Medical University Vienna. Cancers (Basel). 2020 Nov 26;12(12):3533. doi: 10.3390/cancers12123533. 
  • Mladenova I. Helicobacter pylori and cardiovascular disease: update 2019. Minerva Cardioangiol. 2019 Oct;67(5):425-432. doi: 10.23736/S0026-4725.19.04986-7
  • Shindler-Itskovitch T, Chodick G, Shalev V, Muhsen K. Helicobacter pylori infection and prevalence of stroke. Helicobacter. 2019 Feb;24(1):e12553. doi: 10.1111/hel.12553. 
  • Li JZ, Li JY, Wu TF, Xu JH, Huang CZ, Cheng D, Chen QK, Yu T. Helicobacter pylori Infection Is Associated with Type 2 Diabetes, Not Type 1 Diabetes: An Updated Meta-Analysis. Gastroenterol Res Pract. 2017;2017:5715403. doi: 10.1155/2017/5715403.

Answer: We thank the reviewer for his/her suggestion and appreciation of our work. We modified the revised version of the manuscript according to reviewer’s comment: “While remaining a local pathogen, H. pylori may exert systemic effects and contribute to the occurrence of clinical extra-gastric manifestations. H. pylori infection, in fact, has been reported to increase the risk of iron deficiency anemia, as well as neurological, cardio-vascular, dermatological and metabolic disorders (Charitos et al., 2021; Durazzo et al., 2021; Mladenova 2019; Li et al., 2017)”. Please see page 1, lines 38-41.

Don't use italics for cytokines' names.

Answer: We thank the reviewer for his/her recommendation. However, we want to precise that we used italics just for cytokine-encoding genes.

Round 2

Reviewer 1 Report

questions addressed